# Temperature Compensation of SAW Winding Tension Sensor Based on PSO-LSSVM Algorithm

**DOI:** 10.3390/mi14112093

**Published:** 2023-11-12

**Authors:** Yang Feng, Wenbo Liu, Haoda Yu, Keyong Hu, Shuifa Sun, Ben Wang

**Affiliations:** 1School of Information Science and Technology, Hangzhou Normal University, Hangzhou 311121, China; fengyang@hznu.edu.cn (Y.F.); liuwenbo@stu.hznu.edu.cn (W.L.); 2023112011031@stu.hznu.edu.cn (H.Y.); watersun@hznu.edu.cn (S.S.); 2School of Engineering, Hangzhou Normal University, Hangzhou 311121, China; hukeyong@yeah.net

**Keywords:** surface acoustic wave (SAW), winding tension sensor, temperature compensation, PSO-LSSVM algorithm

## Abstract

In this paper, a SAW winding tension sensor is designed and data fusion technology is used to improve its measurement accuracy. To design a high-measurement precision SAW winding tension sensor, the unbalanced split-electrode interdigital transducers (IDTs) were used to design the input IDTs and output IDTs, and the electrode-overlap envelope was adopted to design the input IDT. To improve the measurement accuracy of the sensor, the particle swarm optimization-least squares support vector machine (PSO-LSSVM) algorithm was used to compensate for the temperature error. After temperature compensation, the sensitivity temperature coefficient α_s_ of the SAW winding tension sensor was decreased by an order of magnitude, thus significantly improving its measurement accuracy. Finally, the error with actually applied tension was calculated, the same in the LSSVM and PSO-LSSVM. By multiple comparisons of the same sample data set overall, as well as the local accuracy of the forecasted results, which is 5.95%, it is easy to confirm that the output error predicted by the PSO-LSSVM model is 0.50%, much smaller relative to the LSSVM’s 1.42%. As a result, a new way for performing data analysis of the SAW winding tension sensor is provided.

## 1. Introduction

Industrial winding equipment is an essential process of intelligent manufacturing; modern winding equipment with high speed, high precision development, and tension control technology is critical [1]. Tension is a vital evaluation index when monitoring the running state of the winding system. The tension sensor is one of the essential devices for obtaining the tension change of winding equipment, which plays a vital role in improving the performance and parameters of winding capacity equipment [2]. A tension sensor is widely used to monitor the tension of strip, cloth, strip, and linear materials in the production process [3]. In the winding process, it is necessary to keep the winding tension stable to avoid the phenomenon of winding breakage and flaring. 

The precision of winding tension control will directly affect the adaptability of the coil and the quality of the product [4]. Improper tension will affect the transmission effect and production quality of the coil. If the winding tension is too large, the coil is easy to deform or even fracture, affecting the quality of the coil and the appearance of the structure. If the winding tension is too small, it will lead to slippage, lack of form, loose structure, and wrinkles, resulting in rough and uneven rolls, reducing the utilization rate of rolls, etc., thus affecting production quality and efficiency [5].

Tension control is one of the most widely used technologies. In textile yarn winding, M. Ali et al. studied the Arduino Mega 2560 controller to achieve tension control when the yarn is rewound and dyed on a plastic cone [6]. In fiber winding, L. Wen et al. studied a 16-tow prepreg slitting and winding machine, which can automate the place of fiber [7]. In lithium battery aluminum-plastic film winding, Y. Xiao et al. studied the fuzzy PID tension control method of the lithium battery electrode mill based on the Genetic Algorithm (GA) [8]. In slitting machine winding, C. Jiang et al. studied the nonlinear system characteristics of the unwinding system, the unstable diaphragm tension caused by the uncertain interference, and the inaccurate model in the unwinding process [9]. In copper foil winding, I. Jo et al. studied a roll-to-roll (R2R) graphene synthesis system where tension control is crucial to copper foils [10]. In printing and dyeing winding, Ç. Burak et al. studied a method of unwinding and winding textile printing and dyeing machines, which applied to a system of verification of the quality of the textile yarn through image processing [11]. In coil winding, H. Hwang et al. investigated the disturbance observer (DOB) to enhance the robustness of external disturbances to coil mass [12].

The surface acoustic wave (SAW) device has been widely used in mechanical engineering, aeronautics and astronautics, signal processing [13], nondestructive testing, sensor technology, and so on [14], and as important sensing technologies, SAW sensors have exhibited promising characteristics because of their high sensitivity, fast response, excellent specificity, reversibility, battery-powered operation, small size, and low cost, thus offering extensive potential use in the future [15]. The SAW sensors can be used to measure temperature [16], humidity [17], gas [18], force [19], and so on. 

In this paper, we focus on the research of the SAW winding tension sensor. As is known to us, many kinds of sensors can be used to measure tension, such as capacitive tension sensors, resistance tension sensors, hall tension sensors, and so on. However, the output signals of these traditional tension sensors are analog signals, which are susceptible to environmental factors. In contrast, the output signal of the SAW force sensor is a frequency signal, which is less susceptible to interference. In addition, compared to the traditional force sensor, the cost of the SAW winding tension sensor is relatively low. 

However, it cannot ignore the effect of temperature on its measuring accuracy, so to further improve its measurement accuracy, temperature compensation has to be done. This paper used data fusion technology to overcome the shortcomings of the hardware compensation methods. That is, the PSO-LSSVM algorithm was used to compensate for the temperature error of the SAW winding sensor in this paper. We also used the least-squares method to obtain the fitting equation between the output frequency shift and the force of the sensor under different temperature conditions after temperature compensation.

This paper is organized as follows. After this introductory Section 1, the principle of temperature compensation for winding tension sensors by using PSO-LSSVM is shown in Section 2. The SAW winding tension sensor is designed in Section 3. In Section 4, the SAW winding tension sensor is measured by the winding measurement system. Temperature compensation results and analysis are shown in Section 5. This paper’s conclusions are offered in Section 6.

## 2. Principle of Temperature Compensation for SAW Winding Tension Sensor by Using PSO-LSSVM

Figure 1 is the schematic diagram of temperature compensation for the SAW winding tension sensor using PSO-LSSVM. Figure 1 shows the temperature compensator of the SAW winding tension sensor, which is based on data fusion technology to establish a model that can eliminate temperature interference. It mainly consists of the following three parts.

(1)The unwinding roller motor drives the unwinding roller, and the traction roller motor drives the traction roller to work, causing the tension change of the coil. Finally, the surface acoustic wave device changes through the floating roller;(2)The surface acoustic wave winding tension sensor is a delay device composed of an amplifier and feedback circuit. The working environment temperature should be detected by a temperature sensor, which is installed beside the SAW winding tension sensor;(3)The model of eliminating temperature interference of the surface acoustic wave winding tension sensor is based on the data fusion of the PSO-LSSVM model, which finally achieves the purpose of temperature compensation of the sensor.

The output frequency variation Δf of the SAW winding tension sensor and the output $U_{T}$ of the temperature sensor is the input of the temperature compensator, and the compensated result f′ is the output. 

## 3. Design of SAW Winding Tension Sensor 

### 3.1. Design of the Input and Output IDT Using Unbalanced Split-Electrode Interdigital Transducers

The input and output frequency characteristics of the SAW sensor are susceptible to sidelobe interference, which will affect the measurement accuracy of the SAW sensor. The traditional IDT adopts uniform single-electrode interdigital transducers, which means the width of the electrode is equal to the spacing of the electrode, as shown in Figure 2a. However, this design method will cause the IDT to produce a second-order effect [17]. That is, it will make the marginal reflection of the IDT accumulate, thus increasing the sidelobes of the SAW sensor. To suppress the sidelobes, we adopted the unbalanced split-electrode interdigital transducers, namely, in Figure 2b, within each λ length period, each electrode of the input and output IDT is split into two electrodes by 1:3 according to its width a, and then the split-electrodes are arranged at equal intervals. By adopting this structure, the phase of the marginal reflection is close to 180 degrees so that most of the marginal reflection can be offset near the center frequency [19].

As shown in Figure 2b, a and c respectively, the widths of the split-electrode, b and d is the electrode spacing, which satisfies $c=\frac{\lambda }{16},\;e=\frac{3\lambda }{16},and\;b=d=\lambda /8$. The center frequency $f_{0}$ of the sensor design in this paper is 60 MHz, and the selected piezoelectric substrate material is ST-X quartz (S_i_O_2_). The sound wave in this paper is the Rayleigh wave, and its propagation velocity in ST-X quartz is $v_{0}=3158\;m/s$. We have
(1)$$\lambda =\lambda_{0}=\frac{v_{0}}{f_{0}}=\frac{3158}{60000000}=5.263\times 10^{-5}\;m=52.63\;\mu m$$

Thus, we can obtain c=3.290 μm, e=9.869 μm and d=6.579 μm.

### 3.2. Design of the Electrode-Overlap Envelope of the Input IDT

To further suppress the sidelobes of the SAW sensor, the electrode-overlap envelope of the input IDT is weighted according to the Hamming function [20,21], which is a kind of cosine square function, and it is shown as
(2)$$h\left(t\right)=\left[0.08+(1-0.08){cos}^{2}\frac{\pi t}{2\tau }\right]=0.54+0.46cos\frac{\pi t}{\tau }$$
where *τ* is the time length of the input IDT. The electrode-overlap envelope of the input IDT is shown in Figure 2. In addition, the frequency response of Equation (2) is
(3)$$H\left(f\right)=0.54\tau \frac{sin[2\pi f\tau ]}{\pi f\tau }+0.23\tau \left\{\frac{\sin\left[2\pi \left(f\tau -0.5\right)\right]}{\pi \left(f\tau -0.5\right)}+\frac{\sin\left[2\pi \left(f\tau +0.5\right)\right]}{\pi \left(f\tau +0.5\right)}\right\}$$

Equation (3) indicates that 99.96% of the energy is concentrated in the central lobes, and the sidelobes level is extremely low so that it can effectively suppress the sidelobes.

The output transducer is the uniform split transducer. In order to decrease the bulk acoustic wave, the pairs of the electrode number should be more than 20 [22], so we selected the output IDTs finger pairs *N_o_* = 24.

The overall response *H*(*f*) can be split into two summations relating input and output IDTs responses [23], exactly as in Equation (4).
(4)$$H(f)=H_{out}(f)H_{in}(f)$$
where $H_{out}(f)$ is the output IDTs frequency response, and $H_{in}(f)$ is the input IDTs frequency response.

The output IDTs frequency response $H_{out}(f)$ is
(5)$$H_{\mathrm{out}}(\omega )=\sum_{m=1}^{N_{0}}A{\;e}^{-j\omega x_{m}/v}$$
where *A* is the maximum aperture of the IDTs, *x_m_* is the m finger pair of the output finger overlap.

The output IDTs frequency response $H_{in}(f)$ is
(6)$$H_{\mathrm{in}}(\omega )=\sum_{n=1}^{N_{i}}A\;(0.54+0.46cos\frac{\omega x_{n}/v}{2.1})e^{-j\omega x_{n}/v}$$
where *x_n_* is the n finger pair of the output finger overlap.

The −3 dB bandwidth can be written as [24]
(7)$$B=\frac{\Delta f_{-3dB}}{f_{0}}=\frac{\sqrt{2}}{2}\left|H(f)\right|=\frac{\sqrt{2}}{2}\left|H_{in}(\omega )\right|\left|H_{out}(\omega )\right|$$

We selected $N_{o}=24$, so we calculated the relationship between *B* and $N_{i}$*,* that was
(8)$$B=\frac{\Delta f_{-3dB}}{f_{0}}=\frac{6.6\pi }{N_{i}}$$

The bandwidth of the surface acoustic wave device in the winding tension sensor designed in this paper is B=1.43%, then the window length $N_{i}=145$.

To increase the range of the sensor, it is necessary to ensure that the maximum energy of the frequency response is contained within the bandwidth of −3 dB [25]. In this paper, the device bandwidth is controlled by adjusting the window opening time. The window length is defined as *N*, which must be satisfied to ensure that the surface acoustic wave device has a linear phase
(9)$$\tau =\frac{N_{i}-1}{2}$$

Equation (9) is substituted into Equation (2), i.e.,
(10)$$h\left(t\right)=0.54+0.46cos(\frac{2\pi t}{N_{i}-1})$$

Substitute N=145 into the Equation (10)
(11)$$h\left(t\right)=0.54+0.46cos(\frac{2\pi t}{144})$$

Equation (11) is the expression of the function of the envelope curve of the input IDT electrode strip, and t is the propagation time of the surface acoustic wave on the substrate.

The piezoelectric substrate used in the tension sensor is ST-X tangential quartz, and its electromechanical coupling coefficient is only 0.14% [26]. The propagation velocity of the surface acoustic wave on the ST-X tangent quartz substrate $v_{{sio}_{2}}=3158\;m/s$, the center frequency of the designed surface acoustic wave device, $f_{0}=60\;MHz\;m/s$, and the input IDT adopts the unbalanced split-electrode design, and the design parameters are shown in Table 1. Figure 3 shows the SAW transducer mask plate layout by L-edit V 8.30 software. Figure 4 shows the SAW winding tension sensor fabricated on ST-X Quartz substrates.

## 4. Measurement of the SAW Winding Tension Sensor

To the SAW winding tension sensor’s properties, a winding measurement system was developed to explore the association of the input variables with the output variables, as shown in Figure 5. In this measurement system, we applied the network analyzer (E5061 A) to test the SAW winding tension sensor. The network analyzer connected the input and output wires with the testing base. The base consisted of the circuit modules that the SAW winding tension sensor pins to create the path from the sensor to the network analyzer. The frequency characteristics of the SAW winding tension sensor are shown in Figure 6. 

The winding tension exerted on the sensor was between 0 N and 1 N, and the measurement temperature changed from 30 °C to 50 °C. The tension of the winding’s initial value is 0 N, with 0.1 N introduced each time, and the final value is 1 N. The tension value of the winding and the matching frequency difference of the SAW winding tension sensor is recorded. To enrich the sample ability and improve the accuracy of the derived conversion association of the output frequency difference with the winding tension of the SAW winding tension sensor, we increase the amount of testing with the same experimental conditions. Since the output frequency of the SAW winding tension sensor read by a network analyzer is dynamic, the output frequency under the same winding tension ought to select a comparatively stable value. We set the number of tests to 10 and obtained the testing data indicated in Table 2.

In Figure 6, the frequency characteristics of the SAW winding tension sensor perform very well. However, in Figure 7, the measurement results are quite different under different temperatures, which indicates that the temperature variation can seriously affect its measurement accuracy. 

In this paper, the temperature sensitivity coefficient $\alpha_{s}$ measures the effect of temperature on the SAW winding tension sensor. The smaller the temperature sensitivity coefficient $\alpha_{s}$ of the sensor, the stronger its resistance to temperature interference. The temperature sensitivity coefficient *α_s_* can be expressed as
(12)$$\alpha_{s}=\frac{{\Delta d}_{max}}{\Delta T\cdot \Delta f(FS)}$$
where ΔT is the maximum temperature variation range, $\;\Delta d_{max}$ is the maximum deviation of the output frequency shift Δf under all different temperature conditions in the full-range magnitude of the winding tension F, and $\Delta f\left(FS\right)$ is the full-range magnitude of the output frequency shift Δf. The larger the temperature sensitivity coefficient *α_s_*, the greater the influence of temperature on the measurement accuracy of the sensor. As shown in Table 2 and Figure 7, ΔT=50−30=20 °C, Δdmax=16631−12934=3697 Hz (when F=1.0 N), and Δf(FS)=16631 Hz, so
(13)$$\alpha_{s_{0}}=\frac{{\Delta d}_{max}}{\Delta T\cdot \Delta f\left(FS\right)}=\frac{3697}{20\times 16631}=1.1115\times 10^{-2}{\;{}^{\circ}C}^{-1}$$

The result of Equation (13) further verifies that the temperature variations can greatly affect the measurement accuracy of the winding tension sensor. Therefore, to improve the measurement accuracy of the sensor, temperature compensation must be done.

## 5. Temperature Compensation Results and Analysis

The LSSVM algorithm is a supervised learning technique that offers a model designed specifically for addressing regression and classification problems [27]. Compared to the SVM model, LSSVM transforms the quadratic programming problem of SVM into a linear equation-solving problem, thereby increasing computational complexity [28]. This model exhibits exceptional nonlinear modeling capabilities, enabling it to adapt to complex relationships by mapping data into high-dimensional spaces through kernel tricks. Consequently, it aids in predicting the impact of sensor temperature on output. With this advantage, LSSVM can effectively capture complex temperature sensor output relationships in sensor temperature compensation, improving the accuracy of temperature compensation while enhancing prediction accuracy [29]. However, the key parameter regularization parameter *γ* and kernel function parameters *σ* are crucial parameters for the model’s generalization ability and convergence efficiency. The values of these two parameters usually need to be manually adjusted based on specific issues [30].

To sum up, the selection of the above parameters *γ* and *σ* is performed blindly, which increases the likelihood of encountering local optimal solutions. As a global optimization algorithm, the Particle Swarm Optimization (PSO) algorithm possesses robustness, fast convergence rate, and strong global search capability [31]. By employing this algorithm for parameter selection purposes, errors resulting from experiential or random choices can be circumvented while enabling automatic and efficient adjustment of parameters *γ* and *σ* to achieve a globally optimal solution that aligns with the current problem [32]. The parallelism inherent in this algorithm proves highly advantageous for optimizing large-scale datasets and complex models. LSSVM integrates the PSO algorithm into its framework to automatically adapt parameters *γ* and *σ*, enhancing model flexibility across diverse datasets and problems [33,34,35]. This adaptive nature coupled with global optimization properties becomes crucial for sensor temperature compensation since temperature variations can exert intricate non-linear effects on sensor output. Compared to the LSSVM model, PSO-LSSVM can demonstrate better robustness and accuracy. 

For validating the fitting analysis scheme for the SAW winding tension sensor based on the PSO-LSSVM model, this article introduces the PSO algorithm into LSSVM as the selection strategy to improve the effect of the LSSVM model. First, the LSSVM model is trained, and the temperature sensitivity coefficient $\alpha_{s_{1}}$ is calculated. Second, the PSO-LSSVM model is trained, and calculated temperature sensitivity coefficient $\alpha_{s_{2}}$. Finally, the maximum mean error in measurement data, LSSVM curve fitting error, and PSO-LSSVM curve fitting error are calculated. 

### 5.1. Training and Prediction of the LSSVM Model

The Support Vector Machine (SVM) is a machine learning model that converts low-dimensional features of data points into high-dimensional features and uses structural optimization principles to generate decision boundaries. As the most significant contribution, SVM transforms the training of the model into a quadratic programming problem, i.e.,
(14)$$min\frac{1}{2}{\left\|\omega \right\|}^{2}+\frac{1}{2}\gamma \sum_{i=1}^{n}{\xi_{i}}^{2}$$
which has inequality constraints
(15)$$y_{i}\left(\omega^{\tau }\varphi \left(X_{i}\right)+b\right)\geq 1-\xi_{i}$$

In Equation (15), ω represents regression weight; $y_{i}$ represents the $X_{i}$ classification category; γ represents the penalty coefficient; $\xi_{i}$ represents the relaxation factor; $\varphi \left(X_{i}\right)$ is the dimension raising function; b represents the linear regression weight. SVM obtains the decision boundary by solving the quadratic programming problem described in Equations (14) and (15). SVM can effectively solve the problem of linear indivisibility of samples, but its transformed quadratic programming problem has considerable computational complexity.

The Least Squares Support Vector Machine (LSSVM) is an improved method of SVM. LSSVM converts the inequality constraints in SVM into equality constraints, so that the solution of dual problems can be converted into the solution of linear equations, which greatly simplifies the calculation.

In LSSVM, inequality constraints in SVM are transformed into equality constraints, namely
(16)$$y_{i}=\omega^{\tau }\varphi \left(X_{i}\right)+b+\xi_{i},\;i=1,2,\ldots ,n$$

The problem described in the Equations (14) and (15) also can be transformed into
(17)$$L\left(\omega ,b,\xi ,\alpha \right)=\frac{1}{2}\omega^{\tau }\omega +\frac{1}{2}\gamma \sum_{i=1}^{n}\xi_{i}^{2}-\sum_{i=1}^{n}\alpha_{i}\left(\omega^{\tau }\varphi \left(X_{i}\right)+b+\xi_{i}-y_{k}\right)$$
where α  is a Lagrangian operator. Finally, combined with the kernel method, the original linear equation can be transformed into a nonlinear form
(18)$$f\left(X\right)=\sum_{i=1}^{n}\alpha_{i}K\left(X,X_{i}\right)+b$$
where $K\left(X,X_{i}\right)$ represents a kernel function, usually replacing $\varphi \left(X\right)$ and $\varphi \left(X_{i}\right)$. In this paper, the kernel function we used is a radial basis function (RBF) kernel, namely,
(19)$$K\left(X,X_{i}\right)=\exp\left(-\frac{{\left\|X-X_{i}\right\|}^{2}}{2\sigma^{2}}\right)$$
where σ is a fixed parameter of kernel function of $K\left(X,X_{i}\right)$.

The LSSVM model effectively reduces the computational complexity of the model by transforming inequality constraints into equality constraints. But in the LSSVM model, parameter σ, γ , as a fixed parameter, greatly affects the accuracy of the whole model. Therefore, this paper will optimize it through the PSO algorithm to make the LSSVM effect more accurate [27,28,29].

In this paper, the output frequency shift Δf of the SAW winding tension sensor at 34 °C in Table 2 is taken as the expected output value of the LSSVM algorithm, and this algorithm is used to train the output frequency shift Δf under other temperature conditions in Table 3. After temperature compensation, the values of the frequency shift Δf of the SAW winding tension sensor under different temperature conditions were obtained, as shown in Table 3 and Figure 8; ΔT=50−30=20 °C, ${\Delta d}_{max}=10912-10326=586\;Hz$ (when F=0.8 N), and Δf(FS)=10912 Hz, so we have the temperature sensitivity coefficient
(20)$$\alpha_{s1}=\frac{{\Delta d}_{max}}{\Delta T\cdot \Delta f\left(FS\right)}=\frac{586}{20\times 10912}=2.6851\times 10^{-3}\;{{}^{\circ}C}^{-1}$$

### 5.2. Principle and Training of the PSO-LSSVM Model

Particle Swarm Optimization (PSO) is a stochastic optimization algorithm inspired by population behavior. Its core idea is to update each random particle according to the status of other particles and its state in the interpretation space continuously iterates to find the optimal solution. In the idea of the PSO algorithm, the state of each random particle contains location and vector velocity, representing the potential solution and historical update trajectory of the particle. In the process of seeking a solution, each particle will be affected by the global optimal particle and optimal particles in the neighborhood to change its vector speed, and update the position. By continuously iterating the state of the particle, we finally obtain the optimization solutions satisfying given conditions.

Assuming that there are *N* particles in the *D* dimension solution space, the orientation of the i-th particle can be set to a D dimension vector, i.e.,
(21)$$X_{i}=\left(x_{i1},x_{i2},\ldots ,x_{iD}\right),i=1,2,3,\ldots ,N$$

In addition, the vector velocity of the i-th particle can also be set to a *D* dimension vector, i.e.,
(22)$$V_{i}=\left(v_{i1},v_{i2},\ldots ,v_{iD}\right),\;i=1,2,3,\ldots ,N$$

In the update process of the i-th particle, there will be a local particle pole value in the local area, and its orientation can be set to
(23)$$P_{i}=\left(p_{i1},p_{i2},\ldots ,p_{iD}\right),\;i=1,2,3,\ldots ,N\;$$

Each time the particle group is updated, there is a global particle pole value in the solution space, and its orientation can be set to
(24)$$P_{best}=\left(p_{1},p_{2},\ldots ,p_{D}\right)$$

The position and vector velocity of the i-th particle will be updated in the k-th round through the following two equations
(25)$$V_{i}^{k+1}=\omega V_{i}^{k}+c_{1}r_{1}\left(P_{i}^{k}-X_{i}^{k}\right)+c_{2}r_{2}\left(P_{best}^{k}-X_{i}^{k}\right)$$
(26)$$X_{i}^{k+1}=X_{i}^{k}+V_{i}^{k+1}$$

Among them, $r_{1},r_{2}$ is a pseudo-random number of 0 to 1, $c_{1},c_{2}$ is the learning factor, and ω is an inertial coefficient.

This article introduces the PSO algorithm into LSSVM as the selection strategy of parameters σ and γ to improve the effect of the LSSVM model. In the PSO-LSSVM algorithm, we set $c_{1}=1.6,{\;c}_{2}=1.1,\;\varepsilon =10^{-8},\;t_{max}=200,\;\omega_{min}=0.5,\;\mathrm{and}\;\omega_{max}=1.6$. After temperature compensation, the values of the frequency shift Δf of the SAW winding tension sensor under different temperature conditions were obtained, as shown in Table 4 and Figure 9; $\Delta T=50-30=20\;{}^{\circ}C,\;{\Delta d}_{max}=10590-10326=264\;\mathrm{Hz}\;\left(\mathrm{when}\;F=0.8\;N\right)$, and $\Delta f\left(FS\right)=10590\;\mathrm{Hz}$, so we have the temperature sensitivity coefficient
(27)$$\alpha_{s2}=\frac{{\Delta d}_{max}}{\Delta T\cdot \Delta f\left(FS\right)}=\frac{264}{20\times 10590}=1.2466\times 10^{-3}\;{{}^{\circ}C}^{-1}$$

By fitting the above experimental data with the least square method, the fitting curves between the winding tension F and the output frequency shift Δf under different temperature conditions can be obtained, which is shown in Figure 9.

### 5.3. Numerical Comparison

For facilitating data comparison, this paper makes a comparison and analysis of the same data sample set, respectively comparing the unoptimized test data, LSSVM model, and PSO-LSSVM model.

The measurement accuracy is a vital index for surface acoustic wave winding sensors. The output average error is a significant parameter for measuring the accuracy of surface acoustic wave winding sensors. In the range of 0–1 N test interval, the output average error of each tension test point is obtained, that is
(28)$$\sigma_{ae}=\frac{\frac{1}{n}\sum_{i=1}^{n}\left|x_{i}-\bar{x}\right|}{\bar{x}}$$

The average output error is shown in Table 5 and Figure 10. The predicted output error represents the accuracy of the best SAW winding tension sensor. As shown in Figure 10, the average output error of the unoptimized test data is 5.95%. For the LSSVM network, the average fitting output error reaches more than 1.42%, while in the PSO-LSSVM model, the average fitting output error can reach 0.50% or even better. As a result, the PSO-LSSVM model for the temperature compensation of the SAW winding tension sensor is better.

## 6. Conclusions

This paper presented a designed SAW winding tension sensor, and data fusion technology was used to improve its measurement accuracy. To suppress the sidelobes, we choose the unbalanced split-electrode interdigital transducers to design the input and output IDTs. We created the electrode-overlap envelope to design the input IDT. Also, to verify the effect of temperature variations on the sensor, we experimented with the sensors under different temperature conditions by winding the measurement system. We found that the sensor’s frequency characteristics performed very well under the same temperature conditions. However, when the temperature changed, its measurement accuracy was seriously affected. To improve the measurement accuracy, we used the LSSVM and PSO-LSSVM algorithms to compensate for the temperature error. After temperature compensation, the conclusions are as follows.

(1)By using the LSSVM and PSO-LSSVM model, the temperature sensitivity coefficient *α_s_* of the SAW winding tension sensor was decreased from $1.1115\times 10^{-2}\;{{}^{\circ}C}^{-1}$ to $2.6851\times 10^{-3}\;{{}^{\circ}C}^{-1}$ and $1.2466\times 10^{-3}\;{{}^{\circ}C}^{-1}$, which was reduced by an order of magnitude. Thus, data fusion technology can significantly improve the anti-temperature interference performance of the SAW winding tension sensor, thereby further improving its measurement accuracy;(2)The average output error of the unoptimized test data is 5.95%. For the LSSVM network, the average fitting output error reaches more than 1.42%, while in the PSO-LSSVM model, the average fitting output error can reach 0.50% or even better. Thus, the PSO-LSSVM model for the temperature compensation of the SAW winding tension sensor is better.

## Figures and Tables

**Figure 1 micromachines-14-02093-f001:**
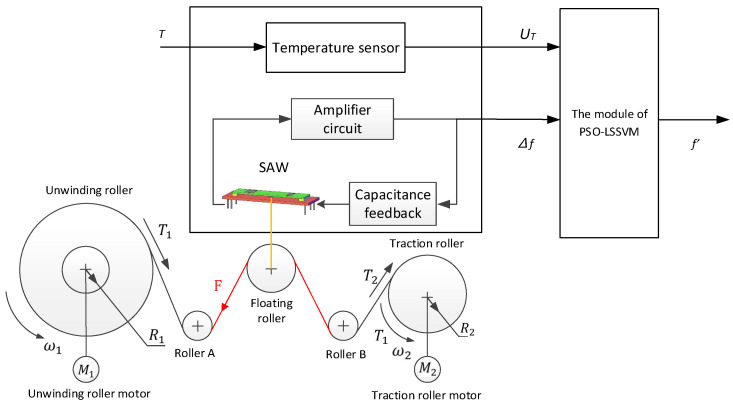
The schematic of the SAW winding tension sensor with differential dual circuit system.

**Figure 2 micromachines-14-02093-f002:**
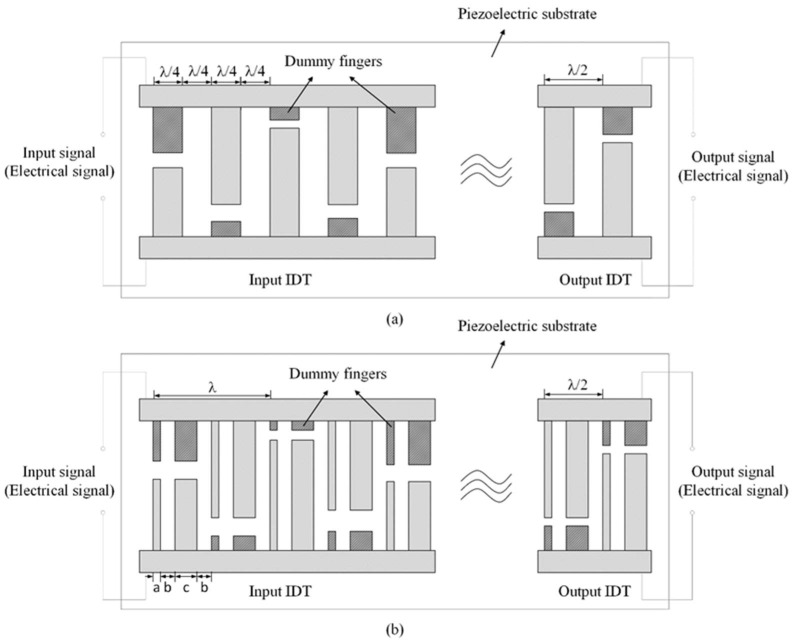
Improved SAW delay-line sensor based on the unbalanced split-electrode interdigital transducers. (**a**) the uniform single-electrode interdigital transducers; (**b**) the unbalanced split-electrode interdigital transducers.

**Figure 3 micromachines-14-02093-f003:**
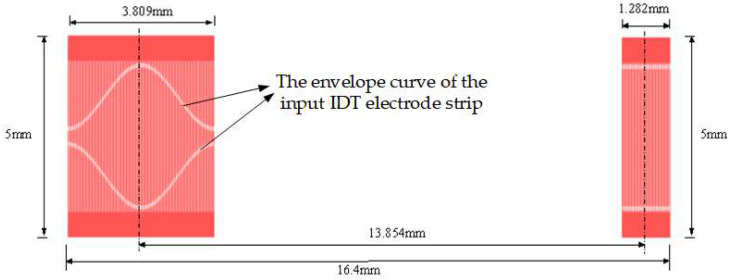
SAW transducer mask plate layout by L-edit software.

**Figure 4 micromachines-14-02093-f004:**
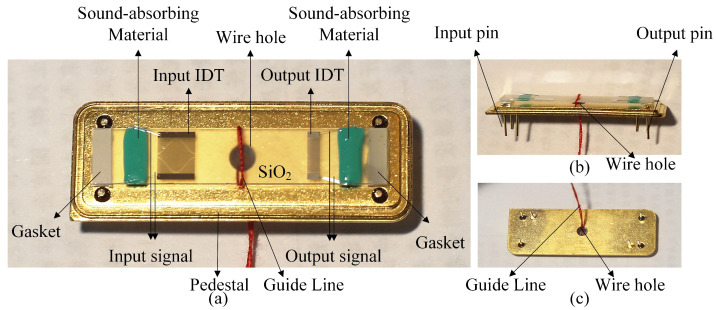
SAW winding tension sensor fabricated on ST-X Quartz substrates. (**a**) the front view; (**b**) the side view; (**c**) the bottom view.

**Figure 5 micromachines-14-02093-f005:**
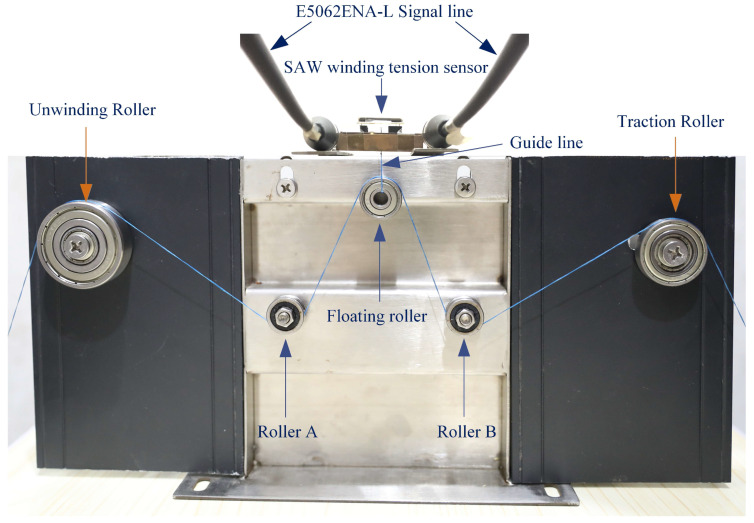
SAW winding tension measurement system.

**Figure 6 micromachines-14-02093-f006:**
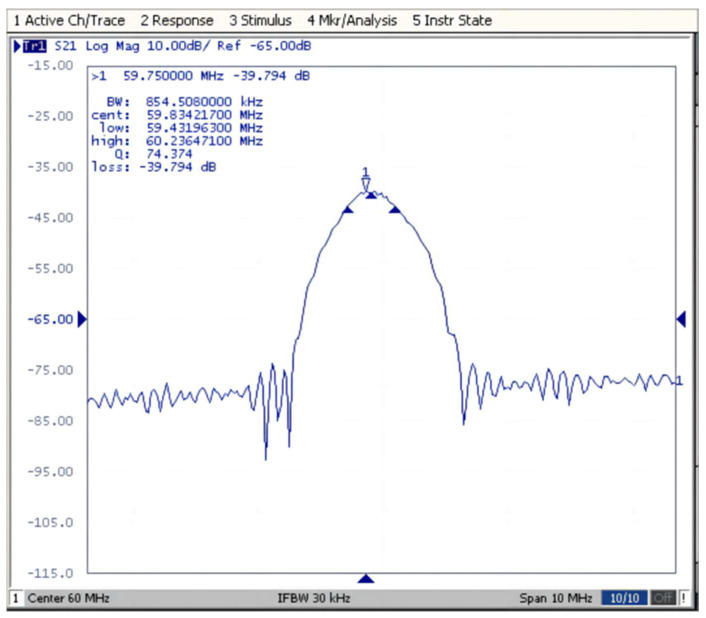
Frequency characteristics of the SAW winding tension sensor.

**Figure 7 micromachines-14-02093-f007:**
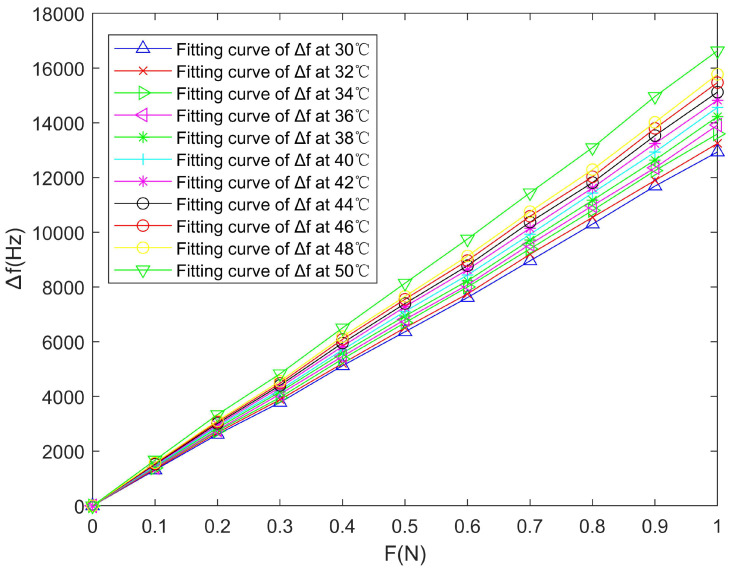
Curve fitting between the tension F and the output frequency shift Δf of the SAW winding tension sensor under different temperatures.

**Figure 8 micromachines-14-02093-f008:**
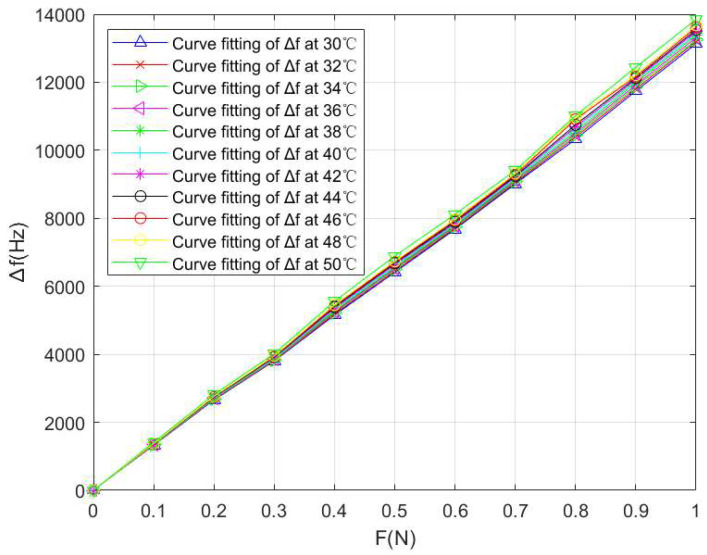
LSSVM curve fitting between the tension *F* and the output frequency shift Δf of the SAW winding tension sensor under different temperatures.

**Figure 9 micromachines-14-02093-f009:**
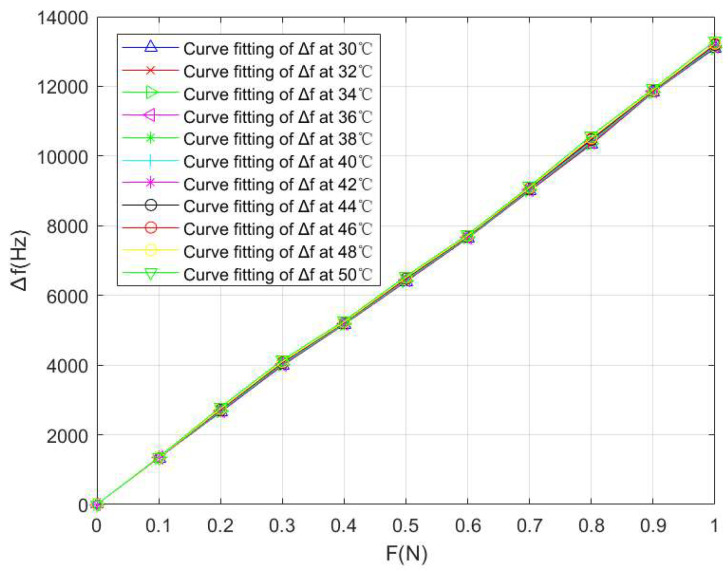
PSO-LSSVM curve fitting between the tension F and the output frequency shift Δ*f* of the SAW winding tension sensor under different temperature conditions.

**Figure 10 micromachines-14-02093-f010:**
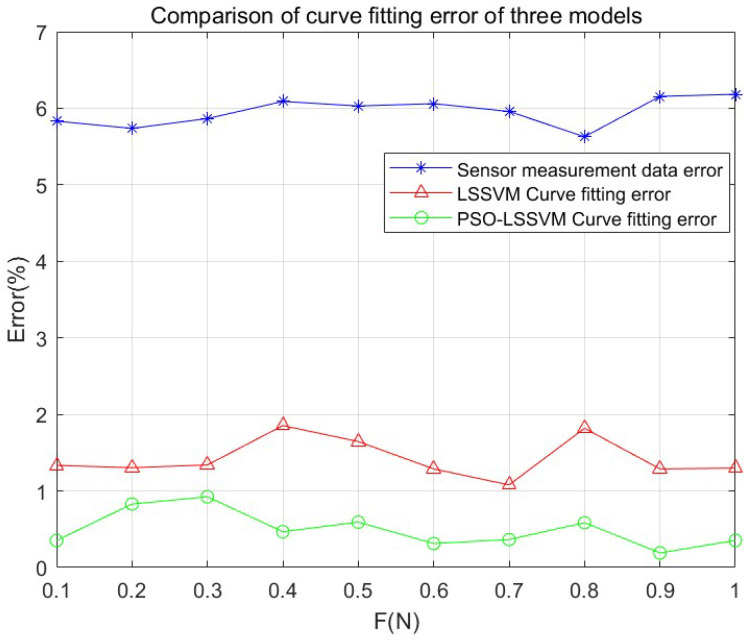
Comparison of the mean error of the SAW winding tension sensor in the 0–1 N measurement interval.

**Table 1 micromachines-14-02093-t001:** Design parameters of SAW winding tension sensor.

Piezoelectric Substrate	Material	ST-X Quartz
	Size	L = 30 mm, W = 6 mm, H = 0.5 mm
IDT	Structure	Delay line
	Frequency of center	60 MHz
	−3 dB bandwidth	1.43% (854.5 Mhz)
	Wavelength Aperture width	λ=52.633333 μm3588.733333 μm
	Number of input IDT	145
	Number of output IDT	49
	Distance between input and output IDT center	27.422 mm (521 λ)

**Table 2 micromachines-14-02093-t002:** Measurement data of the SAW winding tension sensor under different temperatures.

T/°C	*F* (N)										
	0	0.1	0.2	0.3	0.4	0.5	0.6	0.7	0.8	0.9	1.0
	${\Delta f}_{\delta }$ (Hz)										
30	0	1310	2611	3778	5121	6362	7612	8954	10,298	11,687	12,934
32	0	1345	2677	3881	5205	6508	7745	9196	10,542	11,895	13,246
34	0	1385	2738	3960	5367	6694	7999	9391	10,796	12,252	13,595
36	0	1396	2773	4083	5463	6810	8074	9559	10,968	12,356	13,918
38	0	1428	2844	4163	5587	6936	8223	9714	11,168	12,639	14,228
40	0	1451	2905	4232	5701	7105	8435	9922	11,435	12,929	14,564
42	0	1485	2954	4330	5829	7277	8632	10,146	11,619	13,243	14,825
44	0	1518	3011	4403	5951	7393	8778	10,367	11,814	13,532	15,122
46	0	1544	3062	4477	6085	7550	8965	10,583	12,028	13,795	15,471
48	0	1572	3111	4555	6185	7656	9132	10,751	12,288	14,018	15,773
50	0	1671	3336	4826	6503	8138	9752	11,442	13,092	14,963	16,631

**Table 3 micromachines-14-02093-t003:** LSSVM compensation of the SAW winding tension sensor under different temperatures.

T/°C	*F* (N)										
	0	0.1	0.2	0.3	0.4	0.5	0.6	0.7	0.8	0.9	1.0
	${\Delta f}_{\delta }$ (Hz)										
30	0	1326	2657	3804	5152	6411	7677	9012	10,326	11,742	13,132
32	0	1331	2671	3817	5183	6453	7705	9044	10,393	11,801	13,199
34	0	1337	2684	3832	5220	6497	7736	9073	10,449	11,864	13,247
36	0	1342	2698	3845	5246	6529	7765	9109	10,500	11,920	13,313
38	0	1347	2711	3861	5280	6567	7792	9147	10,558	11,988	13,381
40	0	1353	2724	3882	5318	6609	7822	9175	10,632	12,031	13,438
42	0	1358	2734	3900	5349	6642	7855	9212	10,688	12,079	13,497
44	0	1364	2744	3921	5387	6678	7894	9244	10,749	12,123	13,542
46	0	1370	2754	3935	5414	6706	7933	9281	10,912	12,166	13,612
48	0	1375	2763	3953	5453	6736	7968	9320	10,970	12,211	13,679
50	0	1412	2816	4017	5558	6890	8116	9409	11,017	12,439	13,855

**Table 4 micromachines-14-02093-t004:** PSO-LSSVM compensation of the SAW winding tension sensor under different temperature conditions.

T/°C	*F* (N)										
	0	0.1	0.2	0.3	0.4	0.5	0.6	0.7	0.8	0.9	1.0
	${\Delta f}_{\delta }$ (Hz)										
30	0	1334	2648	3986	5166	6397	7641	8996	10,334	11,828	13,096
32	0	1335	2670	4000	5172	6406	7651	9004	10,350	11,836	13,114
34	0	1336	2675	4011	5178	6421	7659	9017	10,375	11,843	13,126
36	0	1337	2684	4023	5187	6434	7668	9026	10,392	11,850	13,138
38	0	1338	2690	4035	5194	6447	7674	9037	10,412	11,856	13,152
40	0	1340	2700	4049	5204	6458	7682	9047	10,432	11,862	13,165
42	0	1342	2714	4063	5213	6472	7691	9059	10,456	11,869	13,180
44	0	1343	2719	40 72	5220	6486	7697	9072	10,471	11,877	13,196
46	0	1344	2721	4086	5226	6500	7705	9084	10,494	11,886	13,215
48	0	1345	2727	4097	5233	6512	7712	9094	10,518	11,895	13,229
50	0	1357	2730	4146	5271	6552	7750	9130	10,590	11,930	13,309

**Table 5 micromachines-14-02093-t005:** The average output error of the SAW winding tension sensor in the 0–1 N measurement interval.

Error (%)	*F* (N)										
	0.1	0.2	0.3	0.4	0.5	0.6	0.7	0.8	0.9	1	Avg
Sensor measurement data error	5.83	5.74	5.87	6.09	6.03	6.06	5.96	5.63	6.16	6.18	5.95
LSSVM Curve fitting error	1.33	1.30	1.34	1.85	1.65	1.29	1.08	1.82	1.29	1.30	1.42
PSO-LSSVM Curve fitting error	0.35	0.83	0.92	0.47	0.59	0.31	0.37	0.59	0.19	0.35	0.50

## Data Availability

Data are contained within the article.

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
