# Peer review of "Temperature Compensation of SAW Winding Tension Sensor Based on PSO-LSSVM Algorithm"

_micromachines, 2023, doi:10.3390/mi14112093_

Round 1

Reviewer 1 Report

Comments and Suggestions for Authors

1.The author should provide the current research status of the Wind Tension Sensor.

2.The author should provide a schematic diagram of how the Wind Tension is applied to the sensor.

3.-3dB bandwidth in the Table 1 is wrong.

4. Why is the insertion loss very high in Figure 6.

5. Please explain the significance of LSSVM and PSO-LSSVM models.

Comments on the Quality of English Language

Extensive editing of English language required. 

Reviewer 2 Report

Comments and Suggestions for Authors

In this article, authors designed SAW based winding tension sensor and employed data fusion technology to achieve refinement in measurement accuracy. Speed, precision and consistency are three most important features of a good industrial winding equipment from manufacturing point of view. To maintain the upkeep of the instrument and maintaining the stability of the winding process, the tension change needs to monitored and regulated. The basis of SAW design aimed towards overcome/improve temperature measurement (in)accuracy. The authors adopted a unique unbalanced split‐electrode IDT design to suppress the sidelobes of the SAW sensor. Authors have developed a winding measurement system to explore the association of the input variables with the output variables. A network analyzer has been applied to test the SAW winding tension sensor. Temperature dependent variation of measurement accuracy has been successfully demonstrated. In order to improve the measurement accuracy of the sensor, temperature compensation must be done. PSO-LSSVM model improves the temperature sensitivity coefficient by applying temperature compensation. This in turn improves accuracy of the SAW winding tension sensor.

Overall, the article presents a comprehensive and well-organized experimental and modeling work. I would encourage the authors to further explore potential practical implications and discuss avenues for future research. This would enhance the overall impact of the article.

Minor comment:

1. Mentioning the sensitivity temperature coefficient by absolute numbers (with 4 decimal places) in the abstract is not necessary. Just mentioning order of magnitude difference should be enough for the abstract.

2. In order to achieve effective suppression of sidelobes, authors used bandwidth B = 1.43%, window length N = 145. How authors come to these specific numbers?? Do authors perform some kind of simulations or optimization to get these specific numbers?

Round 2

Reviewer 1 Report

Comments and Suggestions for Authors

The scientificity and readability of the paper have been greatly improved after modification. In addition, the author should provide detailed modification instructions and explanations.